# An Analysis of ML-Based Outlier Detection from Mobile Phone Trajectories

Francisco Melo Pereira [1,*,†] and Rute C. Sofia [2,†]

1    COPELABS, University Lusofona, 1749-024 Lisbon, Portugal
2    Fortiss GmbH—Research Institute of the Free State of Bavaria for Software Intensive Systems and Services, 80805 Munich, Germany
*    Correspondence: p2809@ismat.pt
†    These authors contributed equally to this work.

**Abstract:** This paper provides an analysis of two machine learning algorithms, density-based spatial clustering of applications with noise (DBSCAN) and the local outlier factor (LOF), applied in the detection of outliers in the context of a continuous framework for the detection of *points of interest (PoI)*. This framework has as input mobile trajectories of users that are continuously fed to the framework in close to real time. Such frameworks are today still in their infancy and highly required in large-scale sensing deployments, e.g., Smart City planning deployments, where individual anonymous trajectories of mobile users can be useful to better develop urban planning. The paper's contributions are twofold. Firstly, the paper provides the functional design for the overall PoI detection framework. Secondly, the paper analyses the performance of DBSCAN and LOF for outlier detection considering two different datasets, a dense and large dataset with over 170 mobile phone-based trajectories and a smaller and sparser dataset, involving 3 users and 36 trajectories. Results achieved show that LOF exhibits the best performance across the different datasets, thus showing better suitability for outlier detection in the context of frameworks that perform PoI detection in close to real time.

**Keywords:** outliers; DBSCAN; LOF; GPS trajectories; machine learning

## 1. Introduction

In the continuous pursuit of Smart City planning and development, municipalities try to find areas of interest for both residents and visitors, in order to better plan available services. Accordingly, it is necessary to find new *points of interest (PoIs)* that truly meet the cultural or even leisure and entertainment needs of the population. In this context, pervasive computing and pervasive technology play a key role, as pervasive technology can support tracking and learning of PoIs in a continuous way, without impact on personal data privacy, and based on the citizen's preferences.

Over the last decade, pervasive technology has been emerging with people-centric hardware and systems and has become more efficient and sustainable. Personal mobile IoT devices [1] such as smartphones are carried around by citizens and can assist in inferring trajectories in an anonymous way, without an impact on privacy. Continuously collected anonymous trajectories represent a set of waypoints. Individual trajectories can then be combined and analysed via different *machine learning (ML)* approaches to provide an estimation of PoIs that relate to the social habits of a population. Related work has been addressing the feasibility and accuracy of relying on pervasive technology as a basis for developing large-scale sensing frameworks that can assist Smart City planning [2,3].

Such frameworks often address planned PoIs by relying on offline data analysis, but not a continuous detection of PoIs over time, e.g., related to seasonal events, or with collective interest, which may change with time. There are, therefore, PoIs that are not implicitly considered so in Smart Cities, and those are usually the ones that are harder to

detect. However, they are highly relevant to achieve efficient city planning, as they are derived from citizens' social behaviour and mobility preferences.

In the traditional definition, a PoI is a point of interest (a location) that may be of interest to someone. Our definition of PoI attempts to capture such locations, currently unknown to a municipality or to a community, and that is revealed to us by the social trajectories of a large number of people, which cross a specific location. The aim of our work is to consider non-intrusive technology to detect such PoIs. To discover them, we use cell phone traces data that reflect the trajectories followed by people in their daily lives. These trajectories reveal their individual stopping points (*stop points*). A cluster in time and in space of stop points can turn out to be a PoI for the local community or for an individual visitor. While trajectories collected from smartphones assist in detecting PoIs, they also include *outliers*. By definition, an outlier is a representation of a point that is beyond the limits of a well-defined population [4]. The traces of stored cell phones, which represent the trajectories followed by users, experience several problems, from communication failures to problems with data recording on the cloud. Hence, an outlier is a cluster of points that derive from an error, and not from stop points. It is this discovery of outliers, in the pursuit of data cleansing, that is reflected in our current work.

In this context, this work proposes a design for a novel non-intrusive framework for inference of PoIs based on mobility trajectories derived from anonymised smartphone data, and performs a performance analysis of DBSCAN and LOF for outlier detection, in the context of a first functional block of such framework, related to outlier detection.

We consider that a relevant framework in this context would be a framework that could assist in inferring similarity in individual (and collective) mobility patterns. Data would be obtained via external devices (e.g., smartphones and other embedded devices carried by citizens), and the inference could be done (a) in real time or (b) close to real time. The initial framework described in this paper considers that trajectory data can be obtained from the city *Internet of Things (IoT)* infrastructure and also from citizens' personal devices, such as smartphones, upon consent.

Such a framework would, therefore, infer some form of human behaviour and map such behaviour in time and space to potential PoIs which can then be deployed on a Smart City dashboard, or on Smart City user applications. To reach such a level of inference, it is necessary to design a framework that can assist a continuous PoI detection, by comparing different ML approaches, and this aspect is currently a major gap in the literature. This work contributes to overcoming such a gap by providing the following contributions:

1.  Provides the initial functional design of a novel framework for continuous detection of PoIs based on smart and anonymised trajectory data collected from personal devices and IoT Smart City infrastructure.
2.  Addresses the issue of outlier detection and provides a validation of outlier detection based on two ML algorithms, *density-based spatial clustering of applications with noise (DBSCAN)* and *local outlier factor (LOF)*.

The first reason for selecting these two specific algorithms, DBSCAN and LOF, lies in the fact that related literature states that these two algorithms are within the ones most relevant in the identification of outliers as shall be debated in Section 2. Some authors, such as Osmar et al., prefer LOF [5]. Other authors, such as Allhussein et al., prefer DBSCAN to analyse outliers [6]. While both algorithms exhibit interesting properties, there is no study comparing both of them in terms of capability to support outlier detection, assuming a framework that relies on trajectory data captured by IoT and personal devices.

A second reason for considering these two specific algorithms lies in the simplicity of both algorithms, a key aspect to consider in a continuous PoI detection framework, which is further addressed in Section 3.

The remainder of this paper is organised as follows. Section 2 provides a description of related literature and of our contributions in comparison to prior work. Section 3 defines our proposal for continuous PoI detection, and its functional blocks, debating PoI detection and inference aspects and introducing also challenges with outlier detection. Section 4

is dedicated to an explanation of which ML algorithms are used to detect outliers giving preference to the most reputable ones and explaining in detail DBSCAN and LOF. Section 5 provides a performance evaluation of the two algorithms based on datasets with distinct features. The paper is summarised in Section 6, where the next steps are also debated.

## 2. Related Work

The use of *mobile crowd sensing (MCS)* [7] applications for urban planning within the context of Smart Cities has been increasing over the last decade. For instance, Yang et al. analyse mobile phone traces to detect specific PoIs such as home and work [2]. The authors show that mobile trajectories can be used to infer specific areas of interest of the user with a fine-grained detail level. However, the detection of outliers is not a core topic in this work. Butron-Revilla et al. address the detection of mobility patterns (and PoIs) based on mobile phone data [8]. Viswanathan et al. focus on situational awareness to define the nature of PoIs (stopovers, specific interest in an area, or occasional stop) [9]. The focus of the authors is on situational awareness and not on semi-automated outlier detection.

Another category of work in regard to PoI detection specifically focuses on outlier detection aspects. The way of approaching the problem of outliers varies from author to author, not only due to different applied methodologies but also due to the different use cases where outlier detection is required.

In this context, Ma et al. rely on the use of LOF for outlier detection in a traffic study [10], primarily using *principal component analysis (PCA)* as an orthogonal linear transformation that transforms the data to a new coordinate system, so that the largest variance by any projection of the data lies along the first coordinate (the so-called first component), the second largest variance lies along the second coordinate, and so on. What the authors aim at is to analyse LOF as an effective and efficient method for outlier detection on data using PCA. Their study is relevant as they analyse the efficacy of LOF based on the variance of coordinates considering different dimensions. However, the efficacy in terms of different datasets and comparison with DBSCAN or any other algorithm is not addressed by the authors.

Alghushairy et al. focus on the difference between global and local outlier detectors giving primacy to LOF [5]. Their study refers to data streams used in big data, analysing both parametric and non-parametric methods, and indicating a new methodology for the use of LOF in data streams. This work corroborates the applicability of LOF to data streams and provides a sound methodology for the use of LOF. However, the work does not compare LOF to other algorithms, such as DBSCAN.

Several authors, such as Markou and Singh [11], Goldstein et al. [12], Patcha et al. [13], and Alimohammadi et al. [14], have published surveys and reviews about outlier detection methods, some of which provide an analytical comparison of the features of existing outlier detection methods. In this context, LOF and DBSCAN are among the most popular solutions for outlier detection.

Sabarish et al. propose *trajectory outlier detection algorithms using boundary (TODB)* [15] and as they state *"The main contribution in this paper is outlier trajectories are identified using boundary method and their classification is done based on the constructed boundary."* The authors rely on a convex hull algorithm for all trajectories, classifying them as being outlier trajectories or not. So, they have to use a boundary, in which the trajectories are inserted. For a continuous assessment, this is not feasible.

In the path of finding outlying sub-trajectories [16], Zhipeng and Dechang propose the *density-based trajectory outlier detection (DBTOD)*, using Hausdorff distance computed in metric spaces, becoming a highly complex method with numerous steps. In our case, we are not interested in eliminating sub-trajectories, but points in sub-trajectories that are outliers.

Similarly, in the case of Youcef and Djenouri, the goal of applying the *group trajectory outlier detection (GTOD)* and the *closed DBSCAN k-nearest (CDkNN-GTOD)* [17] algorithms

is to detect groups of trajectories that can be considered as outliers in reference to most trajectories in a dataset.

Recently, Goodge et al. have proposed a graph neural network method, Lunar, to provide learning within the context of outlier detection, citing LOF and DBSCAN as two key approaches in this context due to their simplicity and proved efficacy [18]. This is an interesting approach, which we expect to analyse in a later phase of the development of this work.

## 3. Framework for Continuous PoI Detection in Smart Cities

### 3.1. Smart Cities and Urban Sensing Background

A key aspect in the development of Smart Cities or Smart Communities relates to the use of data collected via *cyber–physical systems (CPS)* installed in specific IoT infrastructures across the city, or data collected from mobile personal devices, such as smartphones. MCS applications, therefore, rely on available IoT infrastructures and on personal CPS to improve people-centric services provided by Smart Cities. MCS is today applied to a wide range of services that enrich the notion of a Smart City, for instance, monitoring of infrastructures (e.g., energy consumption); increased awareness on social behaviour [19,20]; improvement of traffic patterns [21]; detection of PoIs based on user behaviour and user preferences [22]. The use of MCS based on pervasive, opportunistic sensing [20] needs to be seen as a key component of Smart Cities, where collected data can be used to better plan the city, by incorporating personal preferences of users to planned PoIs. In the related literature, most work focuses on recommendations for PoIs, where PoIs are defined in a static way by municipalities, and PoI detection is considered in recommendation engines, e.g., to provide recommendations about potential itineraries around a city [23]. However, MCS brings in the possibility to detect PoIs in a passive way, derived from data collected from city infrastructures and users' personal devices. Here, the key aspect is not to provide recommendations based on pre-established points; instead, it is to derive PoIs based on the mobility behaviour of users in a city. For this purpose, the next sections start by explaining the proposed architecture and then explain the concept of PoI in the context of our work. Then, debate on how the detection of PoIs and outlier detection can be carried out.

### 3.2. Proposed Framework Functional Blocks

Figure 1 provides a functional illustration for the proposed continuous PoI detection framework.

The presented order of steps is not arbitrary, but some functional blocks may also be placed in a different order. For instance, outliers can be computed before or after the detection of *stop points (SPs)*, i.e., possible PoIs in a trajectory. We chose to perform a prior detection of SPs based on *visit time* (time a user stays stationary in a location), distance travelled, and speed between SPs.

Once trajectory data is received by the framework, such data is analysed to extract parameters that may assist in a fine-grained detection of PoIs, e.g., speed, distance traversed between two readings, and visit time. Additional information may be inferred, e.g., the type of transportation considered by the user may be relevant also to detect similarity in different trajectories and thus assist in the detection of PoIs. The next functional block handles the detection of outliers (B), and afterwards, the data without outliers are used to estimate SPs. While outlier detection may also be handled after estimating SPs, in our opinion, outlier detection is the first step to handle. Before the SP inference, the undesirable values (outliers) must be removed so that no erroneous conclusions can be drawn.

Both the outlier detection and the SP inference require ML application. Therefore, in the context of this paper, we focus on the next sections on the application of ML for outlier detection.

Still, in the context of SP inference, there is the need to validate next the obtained SPs, by removing SPs that may occur sporadically, for instance.

For instance, it is feasible to consider the number of times that the detected SPs are visited, and then cross-reference each SP across different individual trajectories. PoIs can then be obtained after validating the number of different trajectories sharing a common SP, and comparing the average visit time against individual visit times of that SP. Then, a final list of PoIs will be created.

However, this process is continuous, and when repeated for new data sources and respective trajectories, the PoIs already found are automatically removed during the SP detection in order to discover new PoIs.

The next section provides further detail concerning PoI detection, and Section 3.4 provides more detail concerning outlier detection.

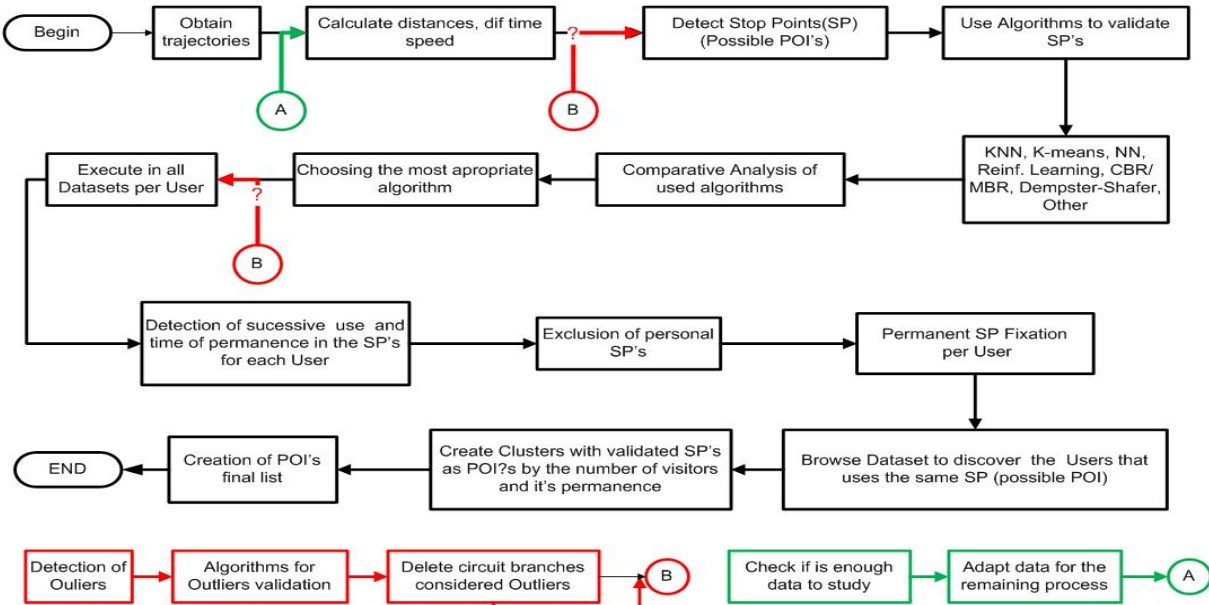

**Figure 1.** Functional blocks of a framework for continuous detection of PoIs in a Smart City, assuming data obtained in close to real time from MCS applications.

### 3.3. Interdisciplinary PoI Definition and Detection Aspects

The definition of PoI [4] in the context of our work relates to MCS and integrates user social behaviour.

Our PoI concept goes beyond the usual spatial data (e.g., GPS coordinates), and follows the line of work that considers PoIs to be a product of space, time, and some measure of influence/attraction [24]. For instance, Chan et al. defined a framework for personalised tour recommendations based on user interests and network visit duration [25]. Their work assumes that there are already pre-established PoIs (municipality data), and the recommendation engine provides a recommendation based on such a PoI set only. However, the overlapping of different individual trajectories can also assist in detecting PoIs which are based on user preferences.

From an individual (one user perspective) a PoI is related to the *social attractiveness* of a user to a specific event or activity, which is defined by different attributes, geo-location being one such attribute. The social attractiveness level varies with time and space and increases with a larger visit to a specific location.

For instance, a person can, in his/her daily routine, stop at a specific location due to having met an acquaintance, or even for curiosity and not necessarily due to an activity or event. This would be a transient SP and should not result in a PoI.

From an aggregated perspective, a PoI can be detected by analysing the similarity of different trajectories in terms of social interaction and similarity patterns in social interaction.

While PoI detection is a well-addressed issue in the context of geo-location systems where large and dense datasets are usually applied, we want to be able to detect PoIs

seamlessly, based on MCS and on individual anonymous trajectories collected by Smart City infrastructures and personal devices. Related literature concerning geo-location and localisation aspects describe several techniques for this purpose, e.g., collecting only geographic data users' paths for specific time windows (location), and collecting data from a triangulation of cellular transmitters (location).

In the context of our work, it is relevant to consider seamless ways to correlate the trajectory data collected in close-to-real-time from different users. Such data may be then correlated with existing PoIs (geo-location) as well.

The detection of new PoIs is, therefore, derived from cluster similarity obtained when considering multiple individual trajectories. For this purpose, our work currently considers two different definitions to detect PoIs.

**Definition 1.** *A PoI is defined by its edge betweenness, i.e., it is defined based on the number of individual trajectories crossing a specific point or within a specific radius. A PoI in this context can be detected via similarity analysis within a specific time and space range.*

**Definition 2.** *A PoI is defined by the clustering derived from the spatial overlap of individual trajectories. A PoI in this context is detected when there are clusters that have high density.*

Both definitions are impacted by other parameters, for instance, speed, visit time, and time granularity.

*Speed (v)*, defined in Equation (1), concerns the average speed that an individual user experiences during his/her periodic routine. In Equation (1), *e* corresponds to the distance travelled (in meters) and *t* corresponds to the time of travel.

$$\delta v = \frac{\delta e}{\delta t} \tag{1}$$

Another possibility to define speed is to consider speed as a result of the displacement between points in a trajectory, as provided in Equation (2).

$$v = \frac{e2 - e1}{\delta t} \tag{2}$$

Across multiple individual trajectories, a similarity analysis not just derived from Definitions 1 and 2, but also incorporating speed on segments of multiple trajectories, can assist in reaching a finer-grained detection of PoIs. For instance, let us assume that a specific point is detected by juxtaposing different individual trajectories at an instant in time t. By considering in addition the speed of different users, it may be feasible to detect whether this is really a PoI.

**Visit time (VT)** corresponds to the time interval in seconds that each user spends on a potential PoI. VT is relevant to define the relevancy of such PoI for a cluster of users. Let us consider a potential PoI detected due to a large number of overlapping trajectories (high edge betweenness). Let us assume that 80% of such trajectories exhibit a low VT (seconds). Then, such a PoI should be disregarded.

**Time granularity** is related to the time scale applied in individual trajectories. Different time granularities associated with different trajectories will also impact the detected PoIs as defined in Definitions 1 and 2.

*3.4. Outlier Detection Aspects*

An outlier is, as explained in Section 1, a representation of a point that is beyond the limits of a well-defined population. The detection of outliers is a first step to be worked upon in our proposed framework, as outliers can lead, in data aggregation, to inaccurate values.

Outliers may occur due to pervasive sensing errors, e.g., some malfunctioning of the equipment or software, or even due to migration of data from user devices to the edge

or cloud. However, they may also occur due to some individual change in the citizens' mobility behaviour.

Outliers require detection and correction through ML algorithms. Outlier detection is the focus of this work and is, therefore, addressed in detail in Section 4.

*3.5. Privacy Preservation and Security Considerations*

MCS requires first of all consent by the user. Then, MCS relies on collected data (e.g., visits to wireless networks) that may be piggybacked and thus impact privacy.

It is understood that sometimes, in order to obtain results about citizens' behaviour, the personal data of the user or the smartphone are identified, which can lead to a privacy breach. While MCS do not necessarily collect personal data, according to the *General Data Protection Regulation (GDPR)* adopted in the European Union, any use of MCS applications needs to be consented to by the users.

A continuous PoI detection framework does not, however, require any personal data. In fact, trajectories sensed by a Smart City IoT infrastructure, or by personal phones are usually anonymous and attributes such as the identifier of the user device (e.g., MAC address) are usually obfuscated.

While the handling of these aspects is carried out on user equipment (data source), a PoI framework detection can also be devised in a privacy-preserving manner, where all datasets are equally treated, so that potentially sensitive data is obfuscated, for instance.

Privacy-preserving mechanisms have been addressed over the last decade, for instance, Liu et al. presented several location privacy models and techniques to perform location data protection and anonymisation [26]. Ali et al. drew attention to privacy issues derived from data aggregation based on data from multiple independent sources. Most protocols and algorithms do not integrate security by design, thus requiring a security adjustment [27].

Kifer et al. propose the "no-free-lunch" theorem, which defines non-privacy as a game, to argue that it is not possible to provide privacy and utility without making assumptions about how the data are generated [28]. Recently, Li et al. have proposed the k-anonymity-based privacy protection for trajectory data [29].

Moreover, several entities provide open data so that researchers can benefit from data that has to be cleaned, because without such a procedure data inevitably brings in significant challenges for the protection of data privacy.

**4. ML for Outlier Detection**

Outlier detection in the context of the framework explained in Section 3.2 is the first functional block to realise the overall framework. If outliers are not removed, then the resulting PoIs will be inconsistent. Such inconsistency may relate to poor data collection, for instance.

Another use of outlier detection is to discover abnormal patterns. We intend to exclude all points and/or trajectory segments that impair trajectory detection.

ML is, therefore, relevant to be considered in this context. The choice of specific ML algorithms requires an approach that best serves the requirements of the specific solution.

ML algorithms relevant to outlier detection can be categorised as follows [30]:

1. **Nearest neighbour-based**. Based on the comparison of the distances between various points and their nearest neighbours [31].
2. **Density-based** [32]. Based on the measurement of the higher or lower density of points in a given region, which has resulted in a new definition of local outliers, with the same principle. Here we consider both LOF and DBSCAN as representative algorithms of this category.
3. **Distance-based** [33]. Defined by measuring the distances from a given point O to other points, resulting in points in its neighbourhood and outliers. Examples are *k-nearest neighbours (kNN)*, k-*means (k-MEANS) clustering*, and *learning vector quantisation (LVQ)*.

Out of the mentioned approaches, this work considers DBSCAN and LOF as the basis for outlier detection in the proposed framework. DBSCAN and LOF are anomaly detection algorithms that use distance to track down the nearest neighbour clusters, based on the k-NN algorithm.

The next subsections explain the two algorithms.

### 4.1. DBSCAN

DBSCAN [34] follows a data clustering approach. Given a set of points in some space, it groups together points that are closely packed together (points with many nearby neighbours), marking as outliers points that are further away in low-density regions (whose nearest neighbours are too far away). Key strengths of DBSCAN, relevant to our work are: (i) it does not require setting up a specific number of clusters to detect; (ii) it can handle clusters of different shapes, sizes, and densities; (iii) it can identify global outliers; (iv) it does not need to have previous knowledge about the number of clusters to form, not even the format of its clusters.

In terms of weaknesses, DBSCAN often misses the detection of varying density clusters and is known to not work well with high dimensional data. For the specific case of outlier detection, none of these disadvantages apply.

DBSCAN requires two parameters as a base for its calculation. Firstly, it requires the size and boundary of its neighbourhood given by eps ($\epsilon$). Two points are considered to be neighbours if the distance between them is less than or equal to eps ($\epsilon$). Secondly, it requires the minimum number of points that the cluster can contain, given as MinPts and described in Equation (3):

$$V\epsilon(x) = y \mid \delta(x,y) \ \leq \ \epsilon \tag{3}$$

where $V\epsilon(x)$ corresponds to the epsilon neighbourhood of a point $x \in R$ given $\delta(x,y)$ which corresponds to the Euclidean distance between two points x and y. Therefore, Equation (3) means mathematically that the neighbourhood of x with an $\epsilon$ radius is formed by all the points y, given that the distance between these two points is less than $\epsilon$.

In Equation (4), the epsilon neighbourhood of point x must contain at least MinPts number of points.

$$V\epsilon(x) \geq MinPts \tag{4}$$

With eps and MinPts, DBSCAN classifies points according to their position in the vicinity of a point [34].

DBSCAN categorises points as follows:

- **Core point**. The point has at least A point is a core point if there are at least MinPts number of points (including the point itself) in its surrounding area with radius eps.
- **Border point**. The point is reachable from a core point and there are less than MinPts number of points (including the point itself) in its surrounding area with radius eps.
- **Outlier point**. The point is not reachable from any core point (and not a core point).

The following concepts are the basis of the DBSCAN algorithm:

- **Direct density reachable:** A point "A" is directly density reachable from another point "B" if:
  1. "A" is in the eps-neighbourhood of "B" and,
  2. "B" is a core point.
- **Density reachable:** A point "A" is density reachable from "B" if there are a set of core points leading from "B" to "A.
- **Density connected:** Two points "A" and "B" are density connected if there is a core point "C", such that both "A" and "B" are density reachable from "C".

### 4.2. LOF

LOF [35] is an algorithm capable of detecting outliers based on the local deviation of a given point in regard to its positioning to its neighbours.

It is, therefore, also a density-based algorithm that shares with DBSCAN the concepts of core distance and reachability distance. LOF can consider nearby local neighbourhoods and not the global distribution of the data. Outliers are detected as points that have a low density in comparison to the rest of the neighbourhood.

LOF is known to perform better when datasets have a variable density (clusters of variable density). LOF depends on how isolated a point is from its neighbours.

LOF is based on the principles, illustrated in Figure 2, namely, the k-distance of an object p from a point o and the k-distance neighbourhood of p, $N_k$, as provided in Equation (5).

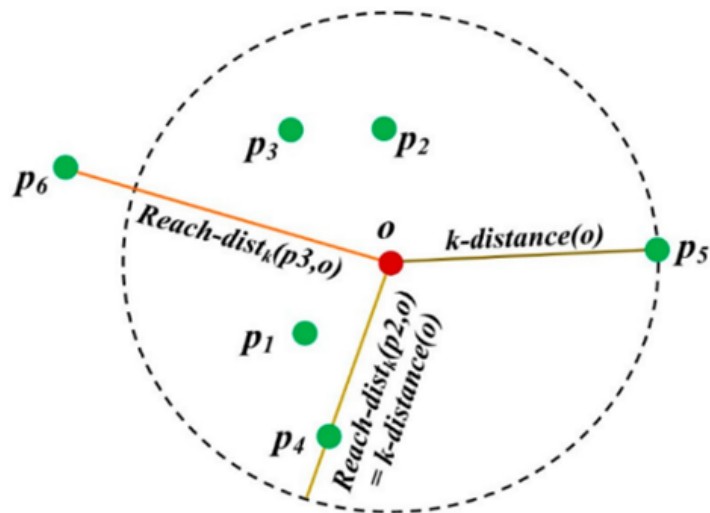

**Figure 2.** LOF principles: k-distance of an object p from a point 0 and k-distance neighbourhood of p, $N_k$, and range distance (rdist) (p).

$$N_{k-distance(p)}(p) = \{q \in D \setminus \{p\} | d(p,q) \leq k - distance(p) \tag{5}$$

$N_{k-distance(p)}$ can be greater than k since several objects can have an identical distance to o. The range distance $rdist(p)$ of p with respect to a point o is provided in Equation (6):

$$rdist(o,p) = maxdist(o,p), kdist(p) \tag{6}$$

Then, the local range density of p is provided in Equation (7):

$$lrd(o) = |R(o)| / (\sum_{p \in R(o)} rdist(o,p)) \tag{7}$$

where:

$$R(o) = \{p | dist(o,p) < kdist(o) \tag{8}$$

The local outlier factor for o is computed via Equation (9):

$$lof(o) = \frac{\sum_{p \in R(o)} lrd(p)}{|R(o)|} / lrd(o) \tag{9}$$

or better applied to our case:

$$LOF_{MinPts}(p) = \frac{\sum_{n \in N_{MinPts}(p)} \frac{Lrd_{MinPts}(o)}{Lrd_{MinPts}(p)}}{|N_{MinPts}(p)|} \tag{10}$$

The lower the local reachability density of o and the higher the local reachability density of o's k-NN, the higher the value of LOF. Given a set of points, LOF computes the potential of each point being an outlier.

LOF is, therefore, capable of capturing local outliers whose local density is relatively low compared to the local densities of its k neighbours.

### 4.3. Additional Algorithm Variants

There are several variants of DBSCAN, such as VDBSCAN, FDBSCAN, DD_DBSCAN, IDBSCAN, OPTCIS, and CRARANS [36], and variants of LOF algorithm, namely, CLOF, LDOF, and NDOT, which serve specific applications dictated by the nature of the project or research in which they are employed [37].

However, in this document, we focus on the real values of DBSCAN and LOF, trying to understand what improvements we can make by varying their parameterisation as well as infer their advantages and disadvantages in finding outliers in trajectories.

### 5. Performance Analysis and Evaluation

This section describes the performance evaluation developed to detect outliers. We have carried out experiments with DBSCAN and LOF for the same datasets.

### 5.1. Methodology

With regard to the methodology used, it can be described as follows:

1. **Selection and analysis of different datasets**. Specific PoI databases are still sparse, in particular considering the parameters proposed for the PoI definition, e.g., speed, visit time, time granularity and geo-location. The dataset selection and cleanup are presented in Section 5.2
2. **Dataset cleanup and validation**. Selected datasets have been transformed into new datasets also integrating time travelled across sequential waypoints of a trajectory; distance travelled; speed; day of the week; user id (obfuscated); means of transport.
3. **Selection of ML algorithms**. The algorithms proposed in this work, DBSCAN and LOF, were applied to detect outliers with the help of Python programming language, proceeding where necessary to any adjustments of parameters or possible tuning of the algorithms.
4. **Performance evaluation**. The accuracy of the algorithms used was measured, verifying their validity and applicability throughout the study, and verifying which factors were preponderant for the analysis.

### 5.2. Datasets

The analysis has been done based on two types of datasets. The first is Geolife (GEO) (https://www.microsoft.com/en-us/download/details.aspx?id=52367, accessed on 12 August 2022 ), provided by Microsoft Research Asia and comprising trajectories involving 178 users from April 2007 to October 2011 across China. Each trajectory is based on a set of GPS-based points ordered in time, comprising longitude, latitude, and altitude. In our study, after an analysis of the different data of GEO, we have randomly selected a total of 380 trajectories from three different users, some of them rejected by the algorithms for having less than twenty points, which is not enough to analyse stop points, and others for not having (according to both algorithms) any outliers.

A second dataset (PTM) has been considered as an example of a smaller, sparser dataset, collected by our team via smartphones with the tool Persense (https://m.apkpure.com/persense-mobile-light/com.senception.persenselight, accessed on 12 August 2022) based on the trajectories of three users in the municipality of Portimão, Algarve, Portugal, over 2 months in 2017, in a total of 36 trajectories. Each trajectory provides latitude, longitude, and speed v.

For the analysis, we have used the sci-kitlearn (https://scikit-learn.org/stable/, accessed on 12 August 2022) Python implementation of DBSCAN and LOF.

### 5.3. Performance Evaluation Parameters

We have evaluated the capability of DBSCAN and LOF in detecting outliers based on the classification parameters *precision* and *recall*. We have also analysed the overall model accuracy.

**Accuracy** provides a measure of the overall performance of the algorithm. It corresponds to all true classifications (true positives, true negatives) over all classified values (positive and negative).

By definition, **precision** corresponds to the number of items correctly labelled as true positives, divided by the total number of elements belonging to the positive class. Precision, defined in Equation (11), provides a measure of how well an algorithm can detect only outliers. Precision is provided as a percentage of the number of outliers in the dataset. The precision is as close to one as the false positives are close to zero.

**Recall**, defined in Equation (12), provides a measure of how well all outliers are identified. The recall is defined as the number of true positives divided by the total number of elements that actually belong to the positive class (i.e., the sum of true positives and false negatives, which are items which were not labelled as belonging to the positive class but should have been). A model that produces no false negatives has a recall of 1.

$$Precision = \frac{TP}{TP + FP} \tag{11}$$

$$Recall = \frac{TP}{TP + FN} \tag{12}$$

where TP—true positives; FP—false positives; TN—true negatives; FN—false negatives.

### 5.4. Outlier Detection with DBSCAN

In the context of this work, DBSCAN is applied to different individual trajectories. This is not a trivial task, as the two parameters eps and MinPts require calibration for each individual trajectory. On our proposed framework for continuous detection of PoIs, this implies that the inference engine will have to use a large range of MinPts and eps, as exemplified in Table 1, that is, with the DBSCAN algorithm, a range of eps parameter values has been tested, and for each, a range of MinPts values has also been tested. For instance, assuming an EPS of 7, then a MinPts value of 2 or of 3 has the same result, it assists in detecting 35 outliers.

An eps of 8 and a MinPts of 6 or 7 results in 39 outliers detected. Varying the MinPts from 8 to 9, while keeping eps at 8, provides changes in the detected outliers. From this simple exercise, the aim is to highlight that there is not a clear selection of the eps and MinPts that results in a better selection of outliers. The values presented in Table 1 represent only a manual selection of the wide range of values tested.

To find a way to circumvent this issue, the first approach considered was to use a wide range of values for the radius of the neighbourhood and the number of points to be checked around the core of a cluster. This idea seemed to overcome the problem of having to select different parameters for each trajectory. However, as explained in the previous paragraph, this approach became difficult due to the variability of the results in relation to the parameters used. Due to these results, we abandoned the previous method and redirected our approach to the methodology proposed by Ester et al. [38] and revisited by Starczewski et al. [39]. For each trajectory the following parameters have been selected:

- MinPts = 2 ∗ DIM, where DIM corresponds to the dataset dimension, i.e., the number of features provided in the dataset. If the dimension is high (more than 5 or 6 features), then MinPts = DIM + 1.
- Using nearest neighbours, the average Euclidean distance between points is calculated.
- After sorting these points, the obtained curve is the basis to fix the eps within the value of the inflection of the curve (knee), which normally occurs above the 95th percentile.

**Table 1.** DBSCAN parameters variation and outlier detection impact.

| Values EPS | Min Samples | Outliers Detected |
|:---:|:---:|:---:|
| 7 | 2 | 35 |
| 7 | 3 | 35 |
| 7 | 4 | 42 |
| 7 | 5 | 42 |
| 7 | 6 | 44 |
| 7 | 7 | 44 |
| 7 | 8 | 47 |
| 7 | 9 | 60 |
| 8 | 2 | 31 |
| 8 | 3 | 33 |
| 8 | 4 | 35 |
| 8 | 5 | 38 |
| 8 | 6 | 39 |
| 8 | 7 | 39 |
| 8 | 8 | 40 |
| 8 | 9 | 42 |

The result of applying this methodology is illustrated in Figure 3.

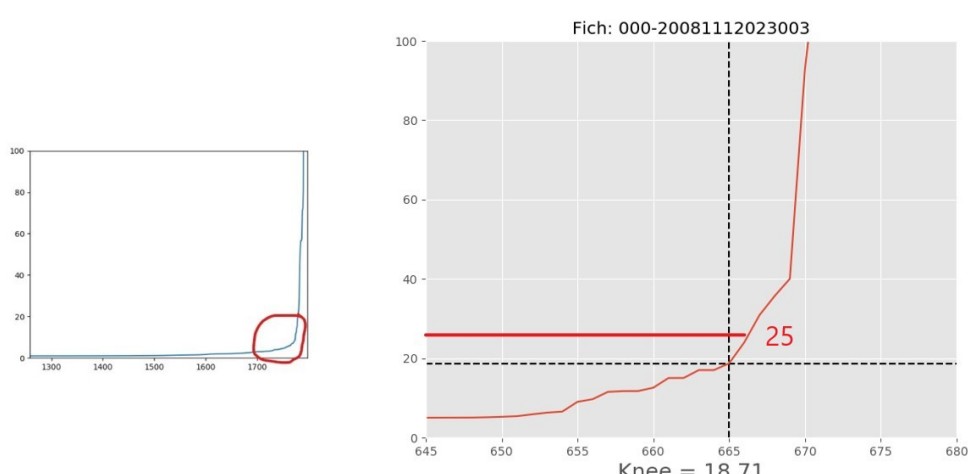

**Figure 3.** DBSCAN knee visualisation example.

This methodology requires human support in the sense that the adaptation requires an analysis of the visual result. This implies an excessive amount of work to be done on each trajectory, an aspect which is not compatible with a continuous detection engine.

Our approach relies on the detection of the knee point, where this point is loosely defined as the point of maximum curvature in a system but also the optimal value in regards to an eps value. This approach is followed, e.g., in Kaggle https://www.kaggle.com/kevinarvai/knee-elbow-point-detection, accessed on 12 August 2022) and better suited for a continuous engine that has to handle multiple trajectories (multiple dataflows).

With the eps value optimised for a chosen MinPts, we provide the calculation with DBSCAN which led us to detect the outliers for each trajectory. A visual representation of

the detected outliers is illustrated in Figure 4 for the two datasets GEO (Geolife) and PTM (Portimão).

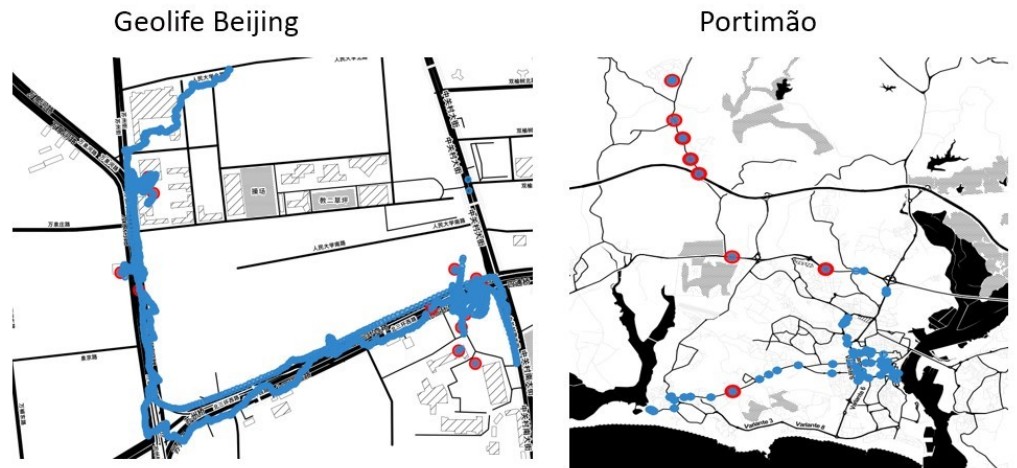

**Figure 4.** Illustration of detected outliers in the GEO and PTM datasets with DBSCAN. Outliers are highlighted in red.

*5.5. Outlier Detection with LOF*

Table 2 illustrates a subset of the computed precision and recall values for some trajectories with LOF. The first column of Table 2 provides the trajectory identifier (named "File Name" in GEO); columns 2 and 3 provide the precision and recall values that have been computed for each trajectory. The number of detected outliers is provided in column 7, while column 8 holds the number of neighbours. For this sample, it can be seen that the values reached for both precisions are usually high (99% or even 100%). Moreover, 0 values mean that the precision or recall could not be computed by LOF.

**Table 2.** Precision and Recall values for LOF.

| Precision | Recall | Accuracy | F-Score | N Reg | Outliers | Neighb |
|---|---|---|---|---|---|---|
| 0.996 | 0.822 | 0.820 | 0.901 | 907 | 6 | 4 |
| 0.000 | 0.000 | 0.000 | 0.000 | 243 | 7 | 1 |
| 1.000 | 0.769 | 0.800 | 0.870 | 49 | 5 | 3 |
| 0.976 | 0.820 | 0.815 | 0.891 | 181 | 13 | 4 |
| 0.000 | 0.000 | 0.000 | 0.000 | 1476 | 67 | 1 |
| 1.000 | 0.876 | 0.877 | 0.934 | 680 | 8 | 11 |
| 1.000 | 0.747 | 0.750 | 0.855 | 493 | 5 | 11 |
| 0.000 | 0.000 | 0.000 | 0.000 | 336 | 23 | 2 |
| 0.985 | 0.817 | 0.824 | 0.893 | 304 | 31 | 3 |
| 0.913 | 0.955 | 0.875 | 0.933 | 80 | 8 | 19 |
| 0.933 | 1.000 | 0.936 | 0.966 | 313 | 32 | 18 |
| 1.000 | 0.250 | 0.333 | 0.400 | 31 | 3 | 3 |
| 1.000 | 0.909 | 0.917 | 0.952 | 80 | 8 | 3 |
| 0.000 | 0.000 | 0.000 | 0.000 | 54 | 6 | 2 |

LOF depends on an adequate calibration of MinPts in order not to have a significant impact on outlier detection.

LOF intrinsically depends on K to determine the scale of local neighbourhoods; however, LOF does not differentiate outliers, and unlike DBSCAN is influenced by the size of the dataset, as can be seen in the correlation provided in Figure 5, which shows a comparison of the dependency of both algorithms on the respective parameters.

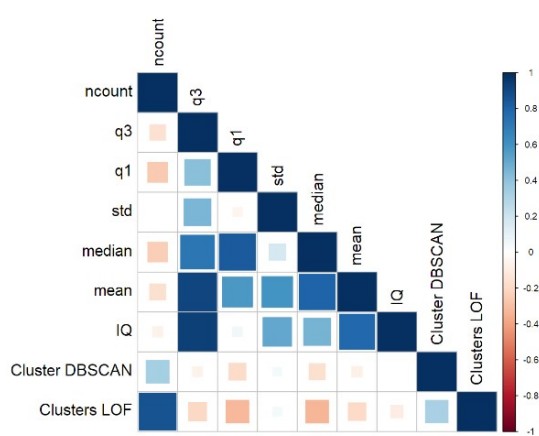

**Figure 5.** Correlation of DBSCAN and LOF to parameters.

Figure 5 shows how the statistical measures (mean, median, quartiles, interquartile distance, and the number of points of each trajectory) are correlated with each other and the clusters obtained through DBSCAN and LOF.

The illustration of the detected outliers with LOF is illustrated in Figure 6.

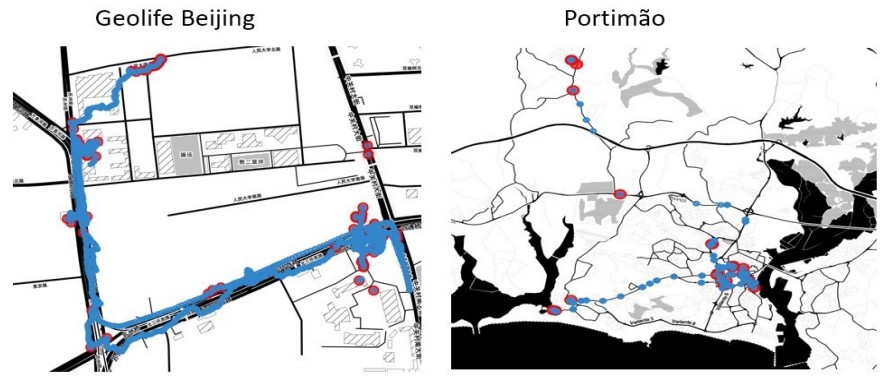

**Figure 6.** Detection of outliers with LOF in the GEO and PTM datasets.

For each point, LOF decides whether it is an outlier or not by checking whether the LOF is close to the value 1. If this value is much higher than 1, it is considered an outlier factor, while if it is close to 1, it is a normal point.

### 5.6. LOF and DBSCAN Comparison

5.6.1. GEO Dataset Results

A global perspective on the performance (precision, recall, accuracy) of LOF, when applied to the GEO dataset, is provided in Figure 7, while the same performance perspective for DBSCAN is provided in Figure 8. For each chart, the X-axis represents the number of individual trajectories available. LOF (refer to Figure 7) reaches a stable precision across all trajectories, while the accuracy varies still within a good level. The recall (how well outliers are detected) results show, however, that LOF exhibits some difficulty in detecting outliers.

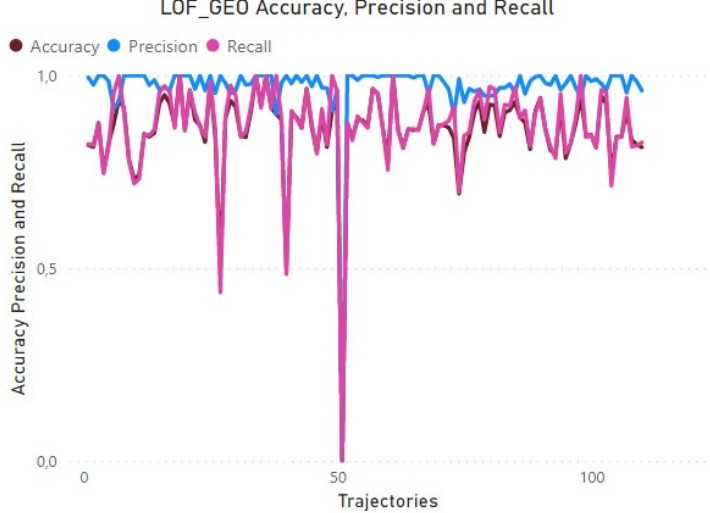

**Figure 7.** LOF performance for the GEO dataset: precision, recall, and accuracy.

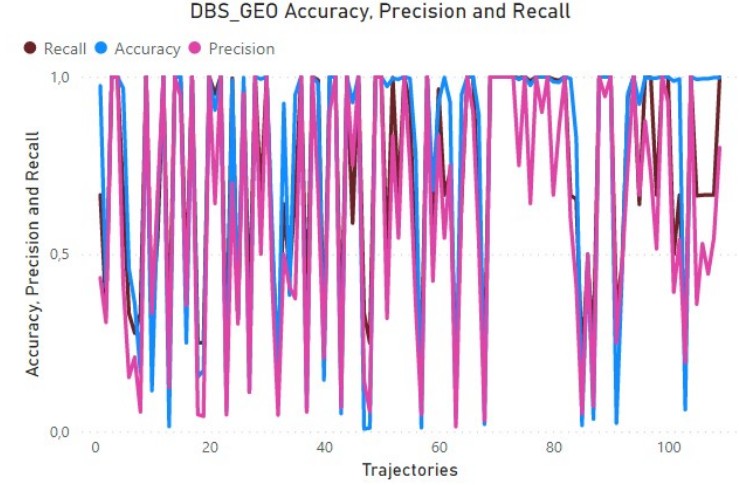

**Figure 8.** DBSCAN performance for the GEO dataset: precision, recall, and accuracy.

DBSCAN (refer to Figure 8) results in more variability in terms of the three evaluation dimensions (precision, accuracy, and recall). Therefore, LOF is the algorithm that performs best in terms of outlier detection for the GEO dataset.

### 5.6.2. PTM Dataset Results

For the smaller dataset, PTM, LOF results are shown in Figure 9 and DBSCAN results are provided in Figure 10. LOF exhibits a good level of accuracy again, but precision and recall are lower.

DBSCAN (refer to Figure 10) has a significantly lower precision, recall, and accuracy.

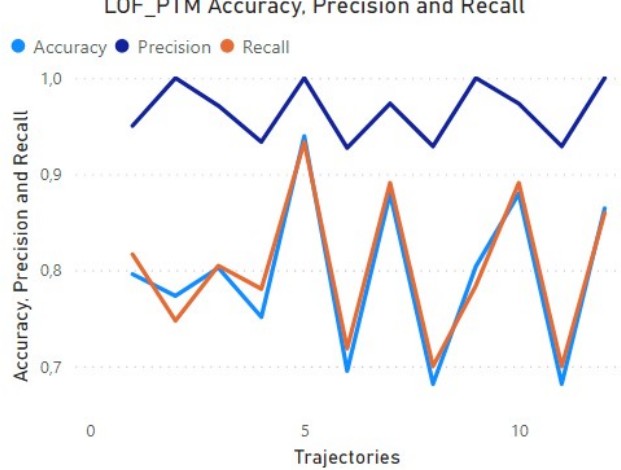

**Figure 9.** LOF performance for the PTM dataset: precision, recall, and accuracy.

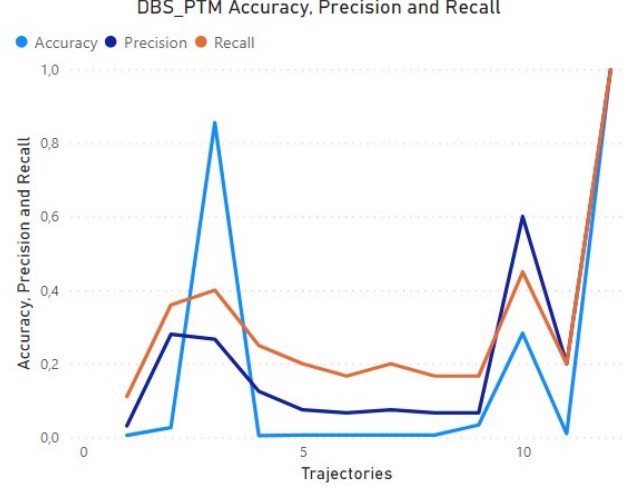

**Figure 10.** DBSCAN performance for the PTM dataset: precision, recall, and accuracy.

*5.7. Discussion of Results*

Table 3 provides the achieved accuracy for LOF and GEO. Overall, LOF provided the best results in terms of outlier detection. The reason for this is related to the fact that LOF gives more importance to local outlier detection than other methods such as DBSCAN.

**Table 3.** Average accuracy of DBSCAN and LOF for the datasets GEO and PTM.

| ML Approach | GEO | PTM |
|:---:|:---:|:---:|
| DBSCAN | 0.76 | 0.19 |
| PTM | 0.86 | 0.80 |

Moreover, LOF is easier to parameterise because it varies by only one parameter (MinPts) and the variability of this factor can be tested more easily. LOF also shows better accuracy on dense datasets (GEO).

DBSCAN is more difficult to parameterise due to the fact that one has to articulate two parameters, eps and MinPts, even if a refinement approach, such as kNN (as we have considered) is applied. DBSCAN exhibited better performance on the sparser dataset (PTM), but overall, lower performance in terms of outlier detection. In the same way, we can compare the performance of DBSCAN in the different datasets, highlighting the fact

that the PTM dataset is very small and, in addition, each trajectory is much smaller and covers a much smaller distance than that revealed by the GEO Dataset.

However, as mentioned, LOF behaves very well in both datasets, showing an accuracy above 80%. While DBSCAN exhibits a more variable behaviour, as accuracy significantly lowers when DBSCAN is applied to a sparse dataset such as PTM.

## 6. Summary and Next Steps

This paper presents an innovative framework for the continuous detection of PoIs based on mobile phone trajectories, and analyses ML-based algorithms, specifically, DB-SCAN and LOF, to be applied for the continuous detection of outliers. The detection of outliers corresponds to one of the relevant functional blocks in the proposed PoI detection framework. To the best of our knowledge and as corroborated in Section 2, where we have analysed related work, the framework for continuous detection of PoIs is novel and presents the basis for a much-desired aspect in urban planning in Smart Cities. This is the possibility to improve services via existing data, via a consented, non-intrusive, and pervasive data-collecting approach. In addition to the architectural design of such a framework, the paper focuses on the detection of outliers. After checking different algorithms as explained in Section 4, DBSCAN and LOF have been selected as they are representative algorithms for both density-based outlier detection and distance-based outlier detection. The aim was to understand which algorithm could best suit a continuous PoI detection framework in the context of outlier detection (one of the proposed blocks).

DBSCAN and LOF have been applied to two datasets with very different characteristics in terms of universe, size, and location features. The paper explains how to best set DBSCAN and LOF, and performs an analysis for accuracy, precision, and recall. Overall, LOF is the algorithm that seems to be better suited to be used as the basis for the continuous detection of outliers.

As the next steps, we will continue with the development of the proposed framework. Once outliers are removed (by applying LOF), we shall work on the detection of PoIs by integrating learning and correction approaches.

**Author Contributions:** Conceptualization, F.M.P. and R.C.S.; methodology, F.M.P.; software, F.M.P.; validation, F.M.P. and R.C.S.; formal analysis, F.M.P. and R.C.S.; investigation, F.M.P.; resources, F.M.P.; data curation, F.M.P.; writing—original draft preparation, F.M.P.; writing—review and editing, F.M.P. and R.C.S.; visualization, F.M.P.; supervision, R.C.S. All authors have read and agreed to the published version of the manuscript.

**Funding:** This work has been partially funded by the research unit COPELABS, University Lusofona, Lisbon, FCT strategic project COPELABS UID/MULTI/04111/2019.

**Data Availability Statement:** The datasets used in this work can be found here: https://github.com/fjmper/PHD-Outliers, accessed on 12 August 2022.

**Conflicts of Interest:** The authors declare no conflict of interest.

## Abbreviations

The following abbreviations are used in this manuscript:

| | |
|---|---|
| MDPI | Multidisciplinary Digital Publishing Institute |
| DOAJ | Directory of Open Access Journals |
| ML | Machine Learning |
| MCS | Mobile Crowd Sensing |
| LOF | Local Outlier Factor |
| DBSCAN | Density-Based Spatial Clustering of Applications with Noise |

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
