# Peer review of "An Analysis of ML-Based Outlier Detection from Mobile Phone Trajectories"

_futureinternet, doi:10.3390/fi15010004_

Round 1
Reviewer 1 Report (Previous Reviewer 1)
As I have noted in my previous reviews, the work has been significantly improved compared to the first version. I believe that minor corrections can be made at this stage to improve its readability.
- In the introduction, the authors use the phrases PoI and Outlier Detection and Stop Points - the paper should precisely explain the difference between these terms and the meaning of each term in the context of using the results in managing Smart City infrastructure. I realize that indirectly the authors explain these issues in Chapter 3, but it needs to be clearly and vividly explained at the beginning of the paper.
- In the paper, the authors present a new framework for PoI detection and present the Outlier Detection method - necessarily in the conclusion of the paper there must be a juxtaposition of their proposed methods with methods or solutions available in the literature.
In conclusion, I find the paper interesting, but the style of presentation of the results deviates somewhat from my preferences. The authors should supplement the paper in the areas indicated. After these additions, it can be published.
Author Response
As I have noted in my previous reviews, the work has been significantly improved compared to the first version. I believe that minor corrections can be made at this stage to improve its readability.
- In the introduction, the authors use the phrases PoI and Outlier Detection and Stop Points - the paper should precisely explain the difference between these terms and the meaning of each term in the context of using the results in managing Smart City infrastructure. I realize that indirectly the authors explain these issues in Chapter 3, but it needs to be clearly and vividly explained at the beginning of the paper.
A: We thank the reviewer for the comments. We have added the explanation required in the introduction.
- In the paper, the authors present a new framework for PoI detection and present the Outlier Detection method - necessarily in the conclusion of the paper there must be a juxtaposition of their proposed methods with methods or solutions available in the literature.
A: We thank the reviewer for the comments. The section that covers work related to ours is Section 2. There, we explain the key pointers we found that have developed similar work to ours. Moreover, we have improved the conclusions providing a description on the methods, and the status of existing solutions; why we have opted specifically for DBSCAN and LOF in this context.
Reviewer 2 Report (New Reviewer)
General
It is not quite clear to me what constitutes an outlier. If we are just applying a ML approach it would just be an anomalous observation. If this is something like a bathroom or bathroom with changing table then it may be anomalous but not an outlier. Perhaps some clarification might help a general reader.
I would think there is quite a bit of opportunity here to use clustering and segmentation to improve the evaluation of outliers. What if people travel in groups (families) – how is dependence taken into account or antidependence? I suppose this is partly addressed in 3.3
I think there needs to be some comment on the sample size that are actually used. Line 470 indicates that not all the data are used or perhaps I do not understand. In the figures just points are displayed. Is it worth displaying some trajectories?
Line 32 – I do not understand why “recurring” is needed
36 drop “an”
55 – change to “overcoming such a gap”
59 – capitalize Addresses
74 change “on” to “of”
126 – replace the comma with a period ie “. In”
148 - I did not quite understand “plan the city”. Perhaps I am assuming that a use of this is to help a visitor better interact with the elements of the city. I don’t think that the attributes of the city would change but perhaps they would.
166 – it looks like the paper, especially the figures were done in Portuguese then translated as there are numerous instances where there are terms still in Portuguese. I suppose this is ok but if it is easy to change it might be better in English.
164, 167 – looks like track changes was not turned off
212 – hyphenate well-addressed
247 – I am not sure I agree with your definition of outlier as prediction is a bit subjective a word. For example, if you have an outlier and include it in the model then it might influence the model fit and hence predictions. Perhaps using “different from expected” might be better. It seems that an outlier is a point that is too far away from a set of points i.e. it is unlikely that a walker who traveled only short distances would travel as far as they did for a POI. Did you look at the outliers that were detected to determine if they actually were? I assume that the lat, long readings are true. Is that not the case?
397 – longitude, “and” speed
Figures 4 and 6 Might it be useful to explain why the points in the upper left for Portimao are outliers for DBSCAN but not LOF. Does this have to do with the radius?
458 – table 2 is File Name the same as trajectory?
470 – this seems to be the first mention of “independent trajectories”. Where does this come from? If there are dependent trajectories then there seems to be a different approach to evaluation that uses one trajectory to fit parameters and the other to evaluate. Am I missing something?
Figures 7 and 8. Again the number of trajectories does not match what was mentioned in the explanation of the data. Perhaps add a comment to the data section.
Note also the “e” slipped in instead of “and” in the y axis label for figures 7-10
Author Response
It is not quite clear to me what constitutes an outlier. If we are just applying a ML approach it would just be an anomalous observation. If this is something like a bathroom or bathroom with changing table then it may be anomalous but not an outlier. Perhaps some clarification might help a general reader.
A: We thank the reviewer for the comments. In the introduction we have added the definition of PoI and outlier, with citations as well.
I would think there is quite a bit of opportunity here to use clustering and segmentation to improve the evaluation of outliers. What if people travel in groups (families) – how is dependence taken into account or antidependence? I suppose this is partly addressed in 3.3
A: We thank the reviewer for the comments. When a group of x individuals travels, we consider each individual to provide a trajectory, as our assumption is that the proposed framework would have as input datasets collected via smartphones (or other mobile devices carried by the user). Assuming large groups, this dependency would have impact on the cluster detection, as a group travelling shows a similar pattern of movement. In the specific outlier detection (which is the case of this paper, one of the blocks of the proposed framework), the movement of a group is not relevant – all points created by that group would in fact increase the likelihood of a point not being an outlier.
I think there needs to be some comment on the sample size that are actually used. Line 470 indicates that not all the data are used or perhaps I do not understand. In the figures just points are displayed. Is it worth displaying some trajectories?
A: We thank the reviewer for the comment. A trajectory corresponds to a sequential set of points traversed by a user. After line 470 we have added the following content: In our study, after an analysis of the different data of GEO, we have randomly selected a total of 380 trajectories from three different users, some of them rejected by the algorithms for having less than twenty points, which is not enough to analyse stop points, and others for not having (according to both algorithms) any outlier. ” Figure 4 provides an illustration of some trajectories, highlighting in red the detected outliers.
Line 32 – I do not understand why “recurring” is needed
A: We thank the reviewer for the note, we have changed the “recurring” by “relying on”.
36 drop “an”
55 – change to “overcoming such a gap”
59 – capitalize Addresses
74 change “on” to “of” (line 72)
126 – replace the comma with a period ie “. In” Line 123
A: We thank the reviewer for the comments. All changes for lines 36,55, 59, 72 and 123 have been addressed.
148 - I did not quite understand “plan the city”. Perhaps I am assuming that a use of this is to help a visitor better interact with the elements of the city. I don’t think that the attributes of the city would change but perhaps they would.
A: Urban planning usually does not consider a social context. Studying traffic congestion problems, creating new bus routes to places of interest, whether they were related with a seasonal event, or is really a place frequently used in a city, are aspects which can be detected via the help of pervasive technology. We believe, based on prior experience in municipalities, that this type of tooling can not just provide a better interaction with the elements of the city, but in fact assist a better planning, by understanding better where people flow.
166 – it looks like the paper, especially the figures were done in Portuguese then translated as there are numerous instances where there are terms still in Portuguese. I suppose this is ok but if it is easy to change it might be better in English.
A: We thank the reviewer for the comments. It is true, this figure comes from the beginning of the study, it was made in Portuguese and the translation missed some words. We have corrected this.
164, 167 – looks like track changes was not turned off
A: We thank the reviewer for this comment. We have checked this part, but could not detect any incongruence.
212 – hyphenate well-addressed
A: We have corrected the lack of hyphen.
247 – I am not sure I agree with your definition of outlier as prediction is a bit subjective a word. For example, if you have an outlier and include it in the model then it might influence the model fit and hence predictions. Perhaps using “different from expected” might be better. It seems that an outlier is a point that is too far away from a set of points i.e. it is unlikely that a walker who traveled only short distances would travel as far as they did for a POI. Did you look at the outliers that were detected to determine if they actually were? I assume that the lat, long readings are true. Is that not the case?
A: We thank the reviewer for the comment. Indeed, the way this was written, it was misleading.To assist the reader, we introduce in section 1. the definition of PoI and of outlier following related literature. Then, on section 3.4 we rely on the same outlier definition.
In terms of outlier detection yes, we have analysed based on the data collected (manually) whether the detected outliers were really outliers.
397 – longitude, “and” speed
A: We have corrected this part, highlighted in blue.
Figures 4 and 6 Might it be useful to explain why the points in the upper left for Portimao are outliers for DBSCAN but not LOF. Does this have to do with the radius?
A: Not quite the radius, but because the points in dataset are more sparse, as explained in section 5.7.
458 – table 2 is File Name the same as trajectory?
A: Yes, “File Name” corresponds to the name of trajectories that Microsoft uses in Geolife. We have added an explanation for the table, explaining also what “File Name” means.
470 – this seems to be the first mention of “independent trajectories”. Where does this come from? If there are dependent trajectories then there seems to be a different approach to evaluation that uses one trajectory to fit parameters and the other to evaluate. Am I missing something?
A: Trajectories are introduced up front in section 1. Please refer to the explanation of contributions of the paper, item 2. In related work, we also address work that relies on individual trajectories. We have changed the term “independent” by “individual”. There are no dependent trajectories in the sense that each trajectory is a set of waypoints by an individual over time and space.
Figures 7 and 8. Again the number of trajectories does not match what was mentioned in the explanation of the data. Perhaps add a comment to the data section.
A: We have provided more detail into the used data on section 5.2.
Note also the “e” slipped in instead of “and” in the y axis label for figures 7-10
A: We thank the reviewer for the comments the typo is now corrected.
Reviewer 3 Report (New Reviewer)
File attached

Author Response
I suspect I was asked to referee your paper because of my interest in outlier detection. So I started by looking at that material.
I agree with lines 171 - 173. The only problem is that there may be so many outliers that they seriously distort the parameter estimates from your initial ML analysis to the extent that good observations appear as outliers. This is sometimes called ‘swamping’.
A: We thank the reviewer for these precious comments. In this study, maps were made with the points, as you can see in the excerpts shown in figures 4 and 6, but with all the points of each trajectory. Geolife is an extensive dataset, with thousands of trajectories. For this specific paper, we have randomly selected 3 users and a total of 380 trajectories. We have now added more detail on the data in section 5.2. By working with 380 trajectories randomly, we have detected manually which points were outliers – this was our baseline.
You start more detailed discussion of outlier detection in §4. I have problems with the details, for example your (3) and the material above. You need the boundary of a neighbourhood. This is given three representations: a symbol ǫ,an abbreviation ‘eps’ and a name ‘Epsilon’. Why so many?
Your (3) is full of undefined symbols. What are x and y. Are we in a two dimensional space, or are these just two data points? Is δ(x, y) the distance be-tween x and y in some metric? Presumably | is read as ‘given’. Should the right-hand side of the equation read y|{δ(x, y) < ǫ}? So then x and y are in the same cluster, provided (4) holds. What is V ? Earlier in the paper it was velocity. What is ǫ(x)? What are P and ǫ(P )?
In writing a scientific paper you need to define the quantities in the equations and also give a verbal introduction to the implications of the relationship.
“Let δ(x, y) be the Euclidean distance between two points x and y. Then if {δ(x, y) < ǫ} x and y belong to the same cluster, provided (4) holds’ But you need to define V as well, and so on.
A: We thank the reviewer for these comments .We have added an explanation for the V nomenclature used in eq 3 and added the reference which provides the proposed definition. Small v corresponds to speed; capital V epsilon of x corresponds to epsilon-neighborhood of a number x in R.
Thus, the formula V ε(x) = y | {δ(x, y) < ε } means mathematically that the neighbourhood of x with ε radius is formed by all the points y ‘|’ (means such as) the distance between these two points is less than ε. We have added this explanation.
The other formula (4) means that neighbourhood of a point x (we have changed P by x to simplify the terminology) and radius ε must contain at least MinPts number of points.
There is no confusion between (v) that stands for speed (in another context) and V ε(x) that means a neighbourhood of a point x with radius ε.
l.97. Not ‘Mathew’ but ‘Ma’.
A: We thank for the note and have again checked the references.
Reviewer 4 Report (New Reviewer)
The manuscript discusses an interesting topic. However, some important logic steps are missing, and the presentation is not clear, which makes it obscure and difficult to understand.
1. Both algorithms requires the calculation of distance. In section 5.2 it indicates that the dimensions includes longitude, latitude, altitude, speed. However, these dimensions are heterogeneous, how are they normalized before calculating the distance? This also leads to the difficulty of understanding the values of "Values EPS" in Table 1.
2. The the choice of parameters in Table 1 seems not representative.
2.1. In Line 423 it states "have to use a large range of MinPts and eps", however, only two eps values of 7 and 8 are used, is there any specific reason why they are "a large range"?
2.2. The values of "Min Samples" covers different ranges for these two eps (2-9 and 1-4), there is no discussion about the details of these parameter choice.
2.3. When eps=8, the "outlier detected" is insensitive to "Min Samples". When "Min Samples" = 2 or 3, the "outlier detected" is insensitive to "Values EPS". Although the conclusion "have to use a large range of MinPts and eps" may be correct, the data in Table 1 can not well support it.
3. In page 12, Line 453 it states "these values are very high". However, without comparison (possibly with DBSCAN results?) it is hard to make this conclusion.
4. In figure 5.
4.1. It tries to show the correlations between some statistics (q1, q3, median, mean, std, etc.). But the manuscript does not state where these statistics come from.
4.2. The manuscipt does not demonstrate what effective information is extracted from this figure. For example, which pair has high correlation? What can be concluded from this high correlation pair?
5. I checked the GEO dataset and it seems like there is no labels about outliers (correct me if I am wrong). When discussing the precision and recall, how do we get the ground truth labels?
Author Response
The manuscript discusses an interesting topic. However, some important logic steps are missing, and the presentation is not clear, which makes it obscure and difficult to understand.
- Both algorithms requires the calculation of distance. In section 5.2 it indicates that the dimensions includes longitude, latitude, altitude, speed. However, these dimensions are heterogeneous, how are they normalized before calculating the distance? This also leads to the difficulty of understanding the values of "Values EPS" in Table 1.
A: We thank the reviewer for these comments. The dimension of the dataset does not mean that all variables were used in the distance calculation, obviously only latitude and longitude were used in both datasets. Speed was used only to calculate Stop Points, but not in the context of this particular problem of finding outliers.
The eps values found in table 1 (small sample of some trajectories) appear there as an example of a first approach to the problem and which indicated that it was not the way to go when choosing eps and minPts values. The explanation that follows the table shows a more consistent way of addressing the problem and that is described in several documents that have been referenced. We have further clarified this aspect in the text.
2.1. In Line 423 it states "have to use a large range of MinPts and eps", however, only two eps values of 7 and 8 are used, is there any specific reason why they are "a large range"?
A: As stated in the last question, there were several values to choose from. Table 1 is just a sample that shows how just a small variation in the choice of eps, brings unseen variation in the number of outliers detected. However, the indicated 'knee' value must be calculated independently for each trajectory. WE have adjusted the text, and improved also the table, to reflect better our line of thought.
2.2. The values of "Min Samples" covers different ranges for these two eps (2-9 and 1-4), there is no discussion about the details of these parameter choice.
- We thank the reviewer for the comment. We have provided a more detailed explanation for the values in Table 1.
2.3. When eps=8, the "outlier detected" is insensitive to "Min Samples". When "Min Samples" = 2 or 3, the "outlier detected" is insensitive to "Values EPS". Although the conclusion "have to use a large range of MinPts and eps" may be correct, the data in Table 1 can not well support it.
A: We thank the reviewer for the note. We have corrected the table by checking again all values; we have selected a sample that may assist us in explaining the situation better.
- In page 12, Line 453 it states "these values are very high". However, without comparison (possibly with DBSCAN results?) it is hard to make this conclusion.
A: Here, the aim is to explain that in some cases LOF provides a high precision and/or recall, while in others, it cannot compute these values (0 values). The comparison with DBSCAN comes later. We have added additional text making this explanation clearer, and why we have table 2.
Table 2 is a sample of some trajectories as it would be hard, due to space constrains and readability, to show the 380 trajectories used for our work.
- In figure 5
4.1. It tries to show the correlations between some statistics (q1, q3, median, mean, std, etc.). But the manuscript does not state where these statistics come from.
4.2. The manuscipt does not demonstrate what effective information is extracted from this figure. For example, which pair has high correlation? What can be concluded from this high correlation pair?
A: Figure 5 shows how the statistical measures (mean, median, quartiles, interquartile distance and number of points of each trajectory) are correlated with each other and the clusters obtained through DBSCAN and LOF. We have added this additional information in section 5.5, and have moved the figure closer to that explanation. As can be seen from the figure, only the LOF is highly influenced by the number of trajectory points.
- I checked the GEO dataset and it seems like there is no labels about outliers (correct me if I am wrong). When discussing the precision and recall, how do we get the ground truth labels?
A: You are correct. The reason of this work is to detect outliers with this Machine Learning Algorithms. After that calculation, we apply as said in section 5.3 the Performance Evaluation parameters, (namely Accuracy, Precision and Recall) to infer the validity of choice of those methods. This conclusions are also represented in figures 7-10 and in table 3.
Round 2
Reviewer 3 Report (New Reviewer)

Author Response
- we thank the reviewer for the thorough revision. We have again revised the english and corrected typos.
- There were no problems with figures nor tables...
Reviewer 4 Report (New Reviewer)
Thanks for the response, but some important table/figures in this version is not displayed. Can you send a version with all the tables and figures?
Author Response
We are sorry that there was a problem with the pdf. We do not understand what may have occurred, as the document has been worked in overleaf, and we see all pictures and tables correctly. We have now uploaded a clean version, again revised due to typos. Other than that, the changes and justification for the changes was provided in round 2.
Round 3
Reviewer 4 Report (New Reviewer)
Thanks for the response. However comment 5 is not addressed. The original comment is
I checked the GEO dataset and it seems like there is no labels about outliers (correct me if I am wrong). When discussing the precision and recall, how do we get the ground truth labels?
The general discussions about accuracy, precision and recall are fine. But the GEO dataset does not explictly state which data points are outliers. Without these ground truth labels, the calculation for TP, FP, TN and FN are non-trivial in such unsupervised learning situation.
More specifically, for example, in lines 430 "divided by the total number of elements belonging to the positive class". Such "positive class" seems like a ground truth from some data source or inference. If there are any non-trivial method, it must be included in the manuscript.
Author Response
Please see the attachment.

This manuscript is a resubmission of an earlier submission. The following is a list of the peer review reports and author responses from that submission.
Round 1
Reviewer 1 Report
The article discusses a very interesting topic, which can find a wide application in the field of SmartCity systems - especially in the context of the planned widespread implementation of 5G systems. Below I present some general comments that came to my mind after reading the article:
- It is unfortunate that the article is written in such a sloppy manner. It makes the analysis of the text difficult and leaves a not very good impression. I know that many of these shortcomings can be removed at the stage of proofreading, but still many of them require a great deal of interference in the text. For example, in the article there are sample affiliations left, the first author has no affiliation. Figures are of very poor resolution, Figure 2 shows text highlighting (the figure is probably a screenshot). References without url appear in the literature list, e.g., item 5 "[CrossRef]". Perhaps the authors of the text mistakenly did not submit the final version of the text.
- The sheer carelessness in the preparation of the text is also evident in the explanation and description of the issues presented. For example, in line 136 the authors introduce the variable e and t (time?) without explaining their meaning. The same is also the case in other places in the text.
- The literature analysis is done very superficially and does not clearly indicate the research gap and the research questions the authors of the text want to answer.
- Definitely, the framework presented in Figure 1 requires a wider commentary. It would also be worth indicating its advantages in relation to the existing solutions. Similarly in case of figures presenting results (e.g. Figure 2) - they should be provided with wider description explaining the essence of presented issues. On the Y axis in Figure 13 there are units of EURO?
- The obtained results are not compared with solutions known from the literature - which would allow to conclude that under certain conditions the proposed method is better.
- The authors note in the introduction: " ... allows for the tracking and learning of POIs in a continuous way, without impact on personal data privacy, a ...". I did not find in the text a detailed description of how the proposed approach addresses the issue of "impact on personal data privacy".
In summary, the text is written very chaotically without a detailed explanation of the proposed methods and the obtained results. This makes it difficult and even impossible in some cases to analyze the proposed solutions. If the authors expand and improve the text - it can be reviewed again.
Author Response
|
futureinternet-1772371 |
An Analysis of ML-based Outlier Detection from Mobile Phone Trajectories |
R: The article discusses a very interesting topic, which can find a wide application in the field of SmartCity systems - especially in the context of the planned widespread implementation of 5G systems. Below I present some general comments that came to my mind after reading the article:
- It is unfortunate that the article is written in such a sloppy manner. It makes the analysis of the text difficult and leaves a not very good impression. I know that many of these shortcomings can be removed at the stage of proofreading, but still many of them require a great deal of interference in the text. For example, in the article there are sample affiliations left, the first author has no affiliation. Figures are of very poor resolution, Figure 2 shows text highlighting (the figure is probably a screenshot). References without url appear in the literature list, e.g., item 5 "[CrossRef]". Perhaps the authors of the text mistakenly did not submit the final version of the text.
A: We thank the reviewer for the comments. We agree that unfortunately the uploaded version had several typos because by mistake the corresponding author did a mistake in uploading the final pdf. We have done a thorough review. Then, additional, and more significant changes to text have been highlighted in blue.
- we have improved the pictures.
- missing aspects, such as equation numbering, and adequate referencing, has also been revised.
- we have double checked the references again, in addition to having done a thorough revision of all the paper.
R: - The sheer carelessness in the preparation of the text is also evident in the explanation and description of the issues presented. For example, in line 136 the authors introduce the variable e and t (time?) without explaining their meaning. The same is also the case in other places in the text.
A:- we thank the reviewer for the careful revision. We have corrected this part, added specific definitions for PoIs, and have added an explanation on the e and t notation as follows: “In Eq. 1, e corresponds to the distance travelled (in meters) and t corresponds to the time of travel.”
Similar methodology has been followed in regards to all notation, equations, and definitions.
R: - The literature analysis is done very superficially and does not clearly indicate the research gap and the research questions the authors of the text want to answer.
A: - we thank the reviewer for the comment. To further highlight the contributions of our work, we have worked the abstract; the introduction, and also section 2, related work where, after each described related work, contributions of the current work are described.
R: - Definitely, the framework presented in Figure 1 requires a wider commentary. It would also be worth indicating its advantages in relation to the existing solutions.
A:- On section 3 , we have provided more detail concerning the framework, PoI detection and definition.
R: Similarly in case of figures presenting results (e.g. Figure 2) - they should be provided with wider description explaining the essence of presented issues. On the Y axis in Figure 13 there are units of EURO?
A: - all figures have been revised; legends now provide more detail; and the text provides more detail for each figure and table.
R: - The obtained results are not compared with solutions known from the literature - which would allow to conclude that under certain conditions the proposed method is better.
A: - The paper focuses on 2 existing algorithms well known in literature and provide a comparison of both for different datasets. To ensure that this is clear to the reader,text has been added (lines 63-73) explaining the key reasons to opt for LOF and DBSCAN. Moreover, in section 2, the role of LOF and DBSCAN and the gap that we expect to overcome with this work has been further debated.
Moreover, to our knowledge and as explained in related work, ours is a first work attempting to understand how to detect outliers in a continuous way.
R: - The authors note in the introduction: " ... allows for the tracking and learning of POIs in a continuous way, without impact on personal data privacy, a ...". I did not find in the text a detailed description of how the proposed approach addresses the issue of "impact on personal data privacy".
A: - We thank the reviewer for the observation. Data privacy and privacy preserving aspects are a highly relevant aspect in particular for this type of frameworks, which rely on continuous data capture provided by smartphones. This has been added as sub-section 3.6.
R: In summary, the text is written very chaotically without a detailed explanation of the proposed methods and the obtained results. This makes it difficult and even impossible in some cases to analyze the proposed solutions. If the authors expand and improve the text - it can be reviewed again.
A: We thank the reviewer for the observations and comments, which we have addressed as thoroughly as possible. We have expanded the text considering all the reviewers’ comments. A thorough review of the writing has been done.

Reviewer 2 Report
The Abstract in the manuscript-at-hand should have been delineated in a much more categorical manner, i.e., it should briefly highlight the overall domain, the challenges of the domain, and the key challenge addressed by the manuscript-at-hand.
The Introduction needs to be considerably improved too, i.e., the notions of the Smart City and the essence of the Points of Interest (PoI) in the same should be delineated in detail. The same is true for the Related Work, wherein the authors should detail both the pros and cons of the referred Literature – some of this Literature is also pretty outdated, i.e., it is advisable to cite the Literature from within the past three, or at the most, from the past five years.
The Framework for Continuous PoI Detection (presented in Section 3) also looks quite superficial. Both PoI Definition Aspects and the PoI Detection Aspects are also simplistic ones, i.e., they are a common sense understanding and there is nothing novel in the same here.
Both DBSCAN (Section 4.1) and LoF (Section 4.2) are established phenomenon and there is perhaps no need to illustrate their underlying principles, i.e., only a mere reference to both would have been more than enough.
The Performance Evaluation has also been carried out via standard Parameters. Critical Analysis of the Results is also missing.
There are considerable number of issues with the Language of the manuscript-at-hand a a careful proofreading is highly indispensable.
Author Response
|
futureinternet-1772371 |
An Analysis of ML-based Outlier Detection from Mobile Phone Trajectories |
Futureinternet-1772371 – „An Analysis of ML-based Outlier Detection from Mobile Phone Trajectories”
Answers to reviewers
Review 2
Submission Date
R: The Abstract in the manuscript-at-hand should have been delineated in a much more categorical manner, i.e., it should briefly highlight the overall domain, the challenges of the domain, and the key challenge addressed by the manuscript-at-hand.
A: We thank the reviewer for the observations, which enrich the paper. We have provided a revision of the abstract adding (highlighted) text.
R: The Introduction needs to be considerably improved too, i.e., the notions of the Smart City and the essence of the Points of Interest (PoI) in the same should be delineated in detail.
A: We have improved the introduction, explaining better the notion of PoIs within urban planning and also providing a clearer explanation on the need for continuous PoI assessment frameworks. Moreover, we have added section 3.1, which provides background on Smart Cities; improved section 3, by explaining better the proposed framework; and improving the sections (now merged) on PoI definition and PoI detection aspects.
R: The same is true for the Related Work, wherein the authors should detail both the pros and cons of the referred Literature – some of this Literature is also pretty outdated, i.e., it is advisable to cite the Literature from within the past three, or at the most, from the past five years.
A: We thank the reviewer for the comment, We have thoroughly revised the related work, adding additional references since 2021 and explaining the relation to our work, and how our contributions fit to related literature.
R: The Framework for Continuous PoI Detection (presented in Section 3) also looks quite superficial. Both PoI Definition Aspects and the PoI Detection Aspects are also simplistic ones, i.e., they are a common sense understanding and there is nothing novel in the same here.
A: We have added more detail on the proposed framework after Fig 1, section 3. We have improved overall section 3, providing more detail on our proposed PoI definition. The sections on PoI definition and PoI detection have been merged, to provide a better understanding of our line of thought-
R: Both DBSCAN (Section 4.1) and LoF (Section 4.2) are established phenomenon and there is perhaps no need to illustrate their underlying principles, i.e., only a mere reference to both would have been more than enough.
A: - We thank the reviewer for the observation. After a careful revision and also based on additional feedback by the other reviewers, we have kept the basic information describing the main features of DBSCAN and LOF. We believe this does not undermine the reading and overall storyline and is helpful for readers that may not be acquainted with both algorithms.
R: The Performance Evaluation has also been carried out via standard Parameters. Critical Analysis of the Results is also missing.
A: We again thank the reviewer for the observation. The Results section (5) has been fully revised. We have added a section on methodology, and worked figures, tables and respective explanations across sections 5.5 and 5.6. In section 5.6, the explanation of results is now separated into results obtained for the GEO dataset and for the PTM dataset.
The evaluation of results achieved is described in section 5, where section 5.5 describes and provides a critical analysis for each algorithm, and section 5.6 is dedicated to the critical comparative analysis between the performance of both algorithms. We have improved section 5.6, giving more detail into the behaviour observed.
R: There are considerable number of issues with the Language of the manuscript-at-hand a a careful proofreading is highly indispensable.
A: We thank the reviewer for the observations and comments, which we have addressed as thoroughly as possible. We have expanded the text considering all the reviewers comments. A thorough review of the writing has been done.

Reviewer 3 Report
Detection of deviations in data is an extremely important topic today. Depending on the type of data analyzed, their specifics, or their size, various algorithms deal with detecting unusual data better or worse. The authors attempted to detect deviations in data using the two most popular algorithms in literature: DBSCAN and LOF. The work lacks information about the research methodology. It isn't easy to understand the idea of experiments in the current form of work. The authors should work on improving the presentation of methodology and results. Only tests are shown in work, e.g., a table (figure 7), but this does not allow you to draw conclusions about all the authors' research.
In my opinion, work in this form cannot be published. The authors did not try to prepare the paper carefully and understandably. In its current form, the work contains a lot of editorial errors:
- The authors do not keep the consistency in using abbreviations LOF, LoF, POI, PoI, etc.
- The authors do not use number equations and do not refer to them in the text.
- The authors do not explain the elements used in the equations.
- The authors in important fragments, which are known that are not their work, do not refer to literature, e.g., the definition of Core/Border/Outlier Point, etc.
- The bibliography has to be improved and presented uniformly.
- The authors should not conclude the superiority of the LOF over the DBSCAN algorithm after analyzing only two data sets. They should perform experiments on a larger and varied number of data sets
- in the text, there are no references to some bibliography items
- Figure 2 description is probably inadequate for its content
- in patterns on page 6, there are no necessary parentheses
- Figure No. 5 is illegible
- All tables, which are now presented as Figures, should be presented as tables.
- The affiliation and all necessary information about the paper's authors are not given entirely.
Author Response
|
futureinternet-1772371 |
An Analysis of ML-based Outlier Detection from Mobile Phone Trajectories |
Futureinternet-1772371 – „An Analysis of ML-based Outlier Detection from Mobile Phone Trajectories”
Answers to reviewers
R: Detection of deviations in data is an extremely important topic today. Depending on the type of data analyzed, their specifics, or their size, various algorithms deal with detecting unusual data better or worse. The authors attempted to detect deviations in data using the two most popular algorithms in literature: DBSCAN and LOF. The work lacks information about the research methodology. It isn't easy to understand the idea of experiments in the current form of work. The authors should work on improving the presentation of methodology and results. Only tests are shown in work, e.g., a table (figure 7), but this does not allow you to draw conclusions about all the authors' research.
A: We thank the reviewer for the comments. To assist the reader, we have added a new sub-section 5.1 where the applied methodology is described.
R: In my opinion, work in this form cannot be published. The authors did not try to prepare the paper carefully and understandably. In its current form, the work contains a lot of editorial errors:
- The authors do not keep the consistency in using abbreviations LOF, LoF, POI, PoI, etc.
- The authors do not use number equations and do not refer to them in the text.
- The authors do not explain the elements used in the equations.
- The authors in important fragments, which are known that are not their work, do not refer to literature, e.g., the definition of Core/Border/Outlier Point, etc.
- The bibliography has to be improved and presented uniformly.
- The authors should not conclude the superiority of the LOF over the DBSCAN algorithm after analyzing only two data sets. They should perform experiments on a larger and varied number of data sets
- in the text, there are no references to some bibliography items
- Figure 2 description is probably inadequate for its content
- in patterns on page 6, there are no necessary parentheses
- Figure No. 5 is illegible
- All tables, which are now presented as Figures, should be presented as tables.
- The affiliation and all necessary information about the paper's authors are not given entirely.
A: We thank the reviewer and apologize for the basic typos. In fact, the corresponding author has uploaded a pdf which was not the final version. Nonetheless, we have again revised the adequate version, and have carefully addressed all possible inconsistencies.

Round 2
Reviewer 1 Report
The authors have taken into account all my comments. I believe that after editorial corrections the article can be published.
Author Response
The reviser had no comments for this revision
Reviewer 2 Report
Thank you for addressing the recommendations.
Accordingly, the overall quality of the manuscript-at-hand has improved.
Author Response

(The authors gave the same response as above.)

Reviewer 3 Report
Unfortunately, the authors did not apply my recommendation (to review the paper in the context of many typos and editor errors). They did not improve the reference section. They did not extend the experiments to more than two datasets. They did not comment on a given list of my remarks separately. They only commented that they revised the paper and uploaded a new version. This paper is still not well prepared to be published in such an influential journal. Therefore I still recommend not accepting the paper in that form.
Author Response
|
futureinternet-1772371 |
An Analysis of ML-based Outlier Detection from Mobile Phone Trajectories |
Revision 2
Futureinternet-1772371 – „An Analysis of ML-based Outlier Detection from Mobile Phone Trajectories”
Answers to Reviewer 3
- First of all we thank the reviewer for the comments on the two rounds of review and would like to apologize as our answers to the reviewer suggestions possibly were not clear. As most suggestions were similarly addressed by the three reviewers, we have changed the document significantly (refer to the text highlighted in blue); done a thorough revision of typos, inconsistencies. For the first submission, there was an unfortunate mistake on our side: the wrong pdf (without affiliation of the corresponding author) was uploaded.
- For this second round, we have again checked the document in terms of spelling, typos. Moreover, the revised document we presented had answers to all the questions raised by the reviewers, and all the aspects mentioned were reviewed. In this round we provide more objective answers to each point raised by the reviewer.
Comments and Suggestions for Authors Revision 1
R: Detection of deviations in data is an extremely important topic today. Depending on the type of data analyzed, their specifics, or their size, various algorithms deal with detecting unusual data better or worse. The authors attempted to detect deviations in data using the two most popular algorithms in literature: DBSCAN and LOF. The work lacks information about the research methodology. It isn't easy to understand the idea of experiments in the current form of work. The authors should work on improving the presentation of methodology and results. Only tests are shown in work, e.g., a table (figure 7), but this does not allow you to draw conclusions about all the authors' research.
Authors:
- We thank the reviewer for the comments. To assist the reader, we have added a new sub-section 5.1 where the applied methodology is described.
- The full section 5 has been changed, and a critical discussion on achieved results has been added to section 5.7 – text in blue reflects the changes.
- All of the results extracted with the 2 datasets have been provided in the paper as URL (rf. to the section on data sets). Providing multiple tables with all results would negatively impact the paper readability. Nonetheless, the reader can check the full set of results obtained.
R: In my opinion, work in this form cannot be published. The authors did not try to prepare the paper carefully and understandably. In its current form, the work contains a lot of editorial errors:
- The authors do not keep the consistency in using abbreviations LOF, LoF, POI, PoI, etc.
Authors:
- We thank the reviewer for the comments and indeed there was a problem with the first pdf (the last version was not uploaded). Still, we have again re-checked the document; corrected spelling; improved English, nomenclature, terminology and acronyms, among others.
R:The authors do not use number equations and do not refer to them in the text.
Authors:
- This was also corrected on the first round of reviews. All equations are numbered and adequately referenced and explained in text.
R:The authors do not explain the elements used in the equations.
Authors:
- All of the elements used in equations have been explained on the first round of reviews. Moreover, we have added also more objective definitions to our model of a PoI.
R:The authors in important fragments, which are known that are not their work, do not refer to literature, e.g., the definition of Core/Border/Outlier Point, etc.
Authors:
- We thank the reviewer for the comments but as can be evidenced, the literature from which the above references were taken, these are described in [37] , [38] , [39] [40].
R: The bibliography has to be improved and presented uniformly.
Authors:
- We have added the following Relevant Bibliography: [4], [5], [6], [6], [7], [8], [18], [20], [21], [22], [23], [24], [27], [29], [30], [31], [32] all relevant literature from the last three years 2020, 2021 and 2022.
- Moreover, the section on related work (2.) has been fully revised. The introduction has also been improved to best explain contributions and we have also added references to back up our line of though.
R: The authors should not conclude the superiority of the LOF over the DBSCAN algorithm after analyzing only two data sets. They should perform experiments on a larger and varied number of data sets
Authors:
- We thank the reviewer for this comment, however we felt that for the study in question the trajectories represented were both sparse and dense, short and long, representative of the trajectories that are intended to exist in a Smart City. Both Datasets were selected after 1 year of research on potential datasets and refer to the circulation in two cities of different magnitudes, Beijing and Portimão in which the circulation of citizens is perfectly adequate to any project on Smart Cities. Choosing random datasets would undermine the purpose of our global project, reason why these two datasets were carefully chosen. As an initial work, we could not find any additional datasets with the required attributes.
R: in the text, there are no references to some bibliography items
Authors:
- We thank the reviewer for the observation. We have changed in this review round the bibliography reference, now using a bib file, to prevent references from appearing without being cited and debated.
R: Figure 2 description is probably inadequate for its content
Authors:
- We thank the reviewer for the observation. The text on the LOF principles has been revised and the legend now includes more information.
R: in patterns on page 6, there are no necessary parentheses
Authors:
- The typo has been corrected.
R: Figure No. 5 is illegible
Authors:
- All figures have been revised and improved, legends included.
R: All tables, which are now presented as Figures, should be presented as tables.
Authors:
- We now present the tables adequately labelled.
R The affiliation and all necessary information about the paper's authors are not given entirely.
Authors:
- The affiliation is now correct, we thank the reviewer for the comment
- In conclusion:
- We thank the reviewer and apologize for the basic typos. In fact, the corresponding author has uploaded a pdf which was not the final version. Nonetheless, we have again revised the adequate version, and have carefully addressed all possible inconsistencies.
Comments and Suggestions for Authors Revision 2
R: Unfortunately, the authors did not apply my recommendation (to review the paper in the context of many typos and editor errors). They did not improve the reference section. They did not extend the experiments to more than two datasets. They did not comment on a given list of my remarks separately. They only commented that they revised the paper and uploaded a new version. This paper is still not well prepared to be published in such an influential journal. Therefore I still recommend not accepting the paper in that form.
Authors:
- We thank the reviewer for his comment, and you are absolutely right that we have not answered item by item, which we are doing now, however, all inaccuracies, bibliographies were corrected, recommendations followed, not agreeing however with the need, at this moment, in which we intended only to analyse the behaviour of these two algorithms, which, as you mentioned very well, are among the best known and used, facing the detection of outliers in trajectories. On that basis, and as explained above, those trajectories fill the propose of our main project.
